# Chemical, Nutrient and Physicochemical Properties of Brown Seaweed, *Sargassum polycystum* C. Agardh (Phaeophyceae) Collected from Port Dickson, Peninsular Malaysia

**DOI:** 10.3390/molecules26175216

**Published:** 2021-08-27

**Authors:** Muhammad Farhan Nazarudin, Nurul Haziqah Alias, Seentusha Balakrishnan, Wan Nurazween Izatee Wan Hasnan, Nur Amirah Izyan Noor Mazli, Mohd Ihsanuddin Ahmad, Ina-Salwany Md Yasin, Azizul Isha, Mohamed Aliyu-Paiko

**Affiliations:** 1Aquatic Animal Health and Therapeutics Laboratory (AquaHealth), Institute of Bioscience, Universiti Putra Malaysia, Serdang 43400, Malaysia; nhaziqahalias95@gmail.com (N.H.A.); seentusha@gmail.com (S.B.); wannurazween98@gmail.com (W.N.I.W.H.); nuramirahizyan@gmail.com (N.A.I.N.M.); mia.abdullah5311@gmail.com (M.I.A.); salwany@upm.edu.my (I.-S.M.Y.); 2Laboratory of Natural Medicines and Products Research, Institute of Bioscience, Universiti Putra Malaysia, Serdang 43400, Malaysia; azizul_isha@upm.edu.my; 3Biochemistry Department, Ibrahim Badamasi Babangida University (IBBU), Lapai 911101, Nigeria; mo.aleeyu.paiko@gmail.com

**Keywords:** metabolite profile, phytochemical, physicochemical, *Sargassum polycystum*, seaweed

## Abstract

Recent increased interest in seaweed is motivated by attention generated in their bioactive components that have potential applications in the functional food and nutraceutical industries. In the present study, nutritional composition, metabolite profiles, phytochemical screening and physicochemical properties of freeze-dried brown seaweed, *Sargassum polycystum* were evaluated. Results showed that the *S. polycystum* had protein content of 8.65 ± 1.06%, lipid of 3.42 ± 0.01%, carbohydrate of 36.55 ± 1.09% and total dietary fibre content of 2.75 ± 0.58% on dry weight basis. The mineral content of *S. polycystum* including Na, K, Ca, Mg Fe, Se and Mn were 8876.45 ± 0.47, 1711.05 ± 0.07, 1079.75 ± 0.30, 213.85 ± 0.02, 277.6 ± 0.12, 4.70 ± 0.00 and 4.45 ± 0.00 mg 100/g DW, respectively. Total carotenoid, chlorophyll *a* and *b* content in *S. polycystum* were detected at 45.28 ± 1.77, 141.98 ± 1.18 and 111.29 µg/g respectively. The total amino acid content was 74.90 ± 1.45%. The study revealed various secondary metabolites and major constituents of *S. polycystum* fibre to include fucose, mannose, galactose, xylose and rhamnose. The metabolites extracted from the seaweeds comprised n-hexadecanoic acid, 1,2-benzenedicarboxylic acid, mono(2-ethylhexyl) ester, benzenepropanoic acid, 3,5-bis(1,1-dimethylethyl)-4-hydroxy- methyl ester, 1-dodecanol, 3,7,11-trimethyl-, which were the most abundant. The physicochemical properties of *S. polycystum* such as water-holding and swelling capacity were comparable to several commercial fibre-rich products. In conclusion, results of this study indicate that *S. polycystum* is a potential candidate as functional food sources for human consumption and its cultivation needs to be encouraged.

## 1. Introduction

Seaweeds are marine algae and sustainable resources that could be commercially cultivated in the Malaysian ecosystem. This is because Malaysia has a rich biodiversity of seaweeds that are of high nutritional value, some of which have not been previously studied. The potential of seaweeds as alternative sources for high-quality food products, fertilisers, phyco-colloids and cosmetic ingredients for the nutraceutical and pharmaceutical industries has attracted the Malaysian government to focus on increasing their production as alternative to aquaculture. This is envisaged to assist in raising household incomes through foreign exchange earnings, providing jobs, expanding alternative means of livelihoods and building business for commercial investment opportunities [1]. Recent increased interest in seaweed is motivated by the attention generated in their bioactive components that has potential applications in the functional food and nutraceutical industries, which are lucrative and have the incentives to reduce metabolic risk factors such as hyperglycaemia, hypercholesterolemia and hyperlipidaemia [2]. This is principally because seaweeds have been consumed as traditional cuisines in many Asian countries, which has led to their utilisation as a functional food to develop among the western countries.

The fibre components of seaweeds essentially contains structural polysaccharides such as alginate and fucoidan in brown seaweed, carageenan, agar and porphyran rich in red seaweeds and ulvan found in green seaweeds. Seaweeds also contain high amounts of minerals due to their natural marine habitat, where the diversity of the minerals they absorb has been noticed to be wide. Research by various workers on the relationship between the nutritional value of feeds and the cultured species of seaweeds have been demonstrated to reveal that the key factors in nutritional value are the contents of essential amino and fatty acids [3,4]. Consequently, the same ten essential amino acids, comprising the long-chain unsaturated fatty acids of the omega-3 series are important dietary nutrients for fish, to help in the promotion of healthy growth and survival [5]. This is because many aquatic organisms have the ability to synthesise some of these compounds from precursor molecules, in limited quantities [4,6].

Increasingly in recent times, micro- and macro-algae are being marketed as ‘functional foods’ or ‘nutraceuticals’. However, these terms have no legal status in many countries, but they describe foods that contain bioactive compounds or phytochemicals that could benefit human and animal health beyond the role of basic nutrition (e.g., anti-inflammatories, disease prevention) [7,8]. In various studies, the potential nutritional or bioactive contents of different seaweeds have been reported. However, only a few of such studies have focused on the bioavailability of nutrients from, and phytochemicals in algal foods. Scientific experiments and studies have mostly concentrated attention predominantly on brown seaweeds and their derivatives, largely because of their perceived sustainability [9]. Therefore, the objectives of the present study were to evaluate the phytochemical composition, identify nutritionally important metabolites contained and investigate the physicochemical properties of brown seaweed, *S. polycystum* sampled from Port Dickson, Malaysia, as potential functional food resources. The overall aim of the study was to explore potential application of the seaweed species abundant on the coast of Peninsular Malaysia, for utilisation as functional food, in order to promote the commercial cultivation of the species. 

## 2. Results

### 2.1. Proximate Composition

Results determining the proximate composition of brown seaweed, *S. polycystum*, are as shown in Table 1. Although the moisture content was low, recorded as 13.70 ± 0.14% DW, ash content was high at 21.38 ± 0.17% DW. Content of protein in *S. polycystum* was relatively appreciable, at 8.65 ± 1.06% DW, whereas total lipids in the species in the present study was noted at 3.42% DW. Determination of crude fibre and total carbohydrate contents in *S. polycystum* yielded 13.55 and 36.55% DW, respectively.

### 2.2. Contents of Total Carotenoids and Chlorophylls

Results for the major photosynthetic pigments studied in seaweeds are usually presented as content of total carotenoids, chlorophylls *a* and *b*, and expressed as µg/100 g dry weight (DW). The values for total carotenoids, chlorophyll *a* and chlorophyll *b* content determined in the *S. polycystum* are shown in Table 2. From the table, it may be observed that the concentration of total carotenoids measured 45.28 ± 1.77 µg/100 g DW, whereas chlorophyll *a* and chlorophyll *b* contents measured 141.98 ± 1.18 µg/100 g DW and 111.29 µg/100 g DW, respectively.

### 2.3. Mineral Content

Results for measurement of mineral content of *S. polycystum* is presented in Table 3. Sodium, Na (8876.45 mg 100/g DW) was detected as the most abundant element in the seaweed, followed by potassium, K (1711.05 mg 100/g DW), calcium, Ca (1079.75 mg100/g DW) and magnesium, Mg (213.85 mg 100/g DW). The detection of Fe, Se and Mn in *S. polycystum* in the present study were in trace quantities, at 277.60, 4.70 and 4.45 mg 100/g DW, respectively.

### 2.4. Fucose-Containing Sulfated Polysaccharides from S. polycystum

Composition analysis of monosacharides from brown seaweed, *S. polycystum* have been shown to be made up principally of fucose, with minor quantities of other monosaccharides, especially xylose, galactose, mannose and rhamnose, as shown in Table 4. From results of the present study, fucose was the highest monosaccharide (23.00 ± 0.02 µg/mL) in *S. polycystum* compared to rhamnose (0.30 ± 0.17 µg/mL); the lowest. Mannose and galactose were however, measured as 0.45 ± 0.02 and 0.75 ± 0.23 µg/mL, respectively.

### 2.5. Phytochemical Screening

Tests on the methanolic extract of *S. polycystum* revealed it to contain steroids, phenols, tannins, saponins, flavonoids, terpenoids and glycosides, as shown in Table 5. The qualitative test results for these substances confirmed the presence of the different secondary metabolites, as all were positive.

### 2.6. Amino Acid Profiles

Analytical result of amino acids measured in *S. polycystum* is as shown in Table 6, confirming the presence of 15 amino acids with clearly resolved separations. Content of total amino acids detected in the present study measured 74.90 ± 1.45 mg g^−1^ DW. The seaweed sample revealed the presence of all essential amino acids (EAAs) in different proportions, except tryptophan, which was most likely destroyed during acid hydrolysis. The total essential amino acids (EAAs) content in *S. polycystum* measured 37.28%, where the highest essential amino acid was noted to be leucine and the most limiting was methionine, followed by thyrosine. Although the ratio of essential amino acids to total amino acids was approximately 0.50, that for EAAs to NEAAs in the seaweed sample was about 1.00.

### 2.7. Metabolite Profiling of S. polycystum

GC-MS analysis of *S. polycystum* yielded 22 metabolites to be identified. These metabolites are of varied chemical classes and most have been reported to exhibit important biological activities. These identified metabolites also include various aliphatic acids and aromatic compounds in the different samples. The identified metabolites with retention times (RT), peak areas (%) and molecular formula are as shown in Table 7. 

Among those metabolites identified in *S. polycystum* (Table 7), the four most abundant recorded are *n*-hexadecanoic acid (10.35%), 1,2-benzenedicarboxylic acid, mono(2-ethylhexyl) ester (7.69%), benzenepropanoic acid, 3,5-bis(1,1-dimethylethyl)-4-hydroxy- methyl ester (7.03%), 1-dodecanol, 3,7,11-trimethyl-(4.26%). Other important metabolites identified in *S. ilicifolium* are phenol, 3,5-bis(1,1-dimethylethyl)-(6.54%) and 3,7,11,15-tetramethyl-2-hexadecen-1-ol (6.24%), hexadecanoic acid, methyl ester (5.28%), 3,7,11,15-tetramethyl-2-hexadecen-1-ol (2.27%) and Squalene (0.74%). 

### 2.8. Physicochemical Properties

Evaluation of physicochemical properties of *S. polycystum* revealed differences in the various parameters measured based on temperature, as shown in Table 8. At room temperature (±25 °C), swelling capacity, SWC was measured as 10.27 ± 0.25 mL/g DW, which increased slightly when temperature was raised to 37 °C (10.43 ± 0.12 mL/g DW) and returned to the original value noted at room temperature (10.27 ± 0.25 mL/g DW) at the highest temperature of evaluation (80 °C). The water-holding capacity, WHC on the other hand, was measured as 2.76 ± 1.02 g/g DW at room temperature but increased to 4.10 ± 0.84 g/g DW at 37 °C before decreasing to 2.89 ± 0.86 g/g DW and 2.90 ± 0.69 g/g DW at 60 °C and 80 °C, respectively. Oil-holding capacity, OHC of the seaweed was measured as lowest at 80 °C and room temperature, but highest at 60 °C and 37 °C.

## 3. Discussion

Seaweed is highly productive marine organism that is suited to, and capable of growing on non-arable land, to utilise waste materials as sources of nutrients. Seaweed therefore, generally vary from terrestrial plants in morphological and physiological characteristics and so also their chemical compositions, as evident in their appearances. The moisture content of *S. Polycystum* measured in the present study was slightly higher than was previously documented (9.95% DW) [10] in same species collected from Kota Kinabalu (west coast of North Borneo). Furthermore, the ash content is higher, indicating appreciable composition of diverse minerals, compared to that in most land-based plants that are noted to contain ash values ranging from 5 to 10% [11]. Nonetheless, data for ash content of the species reported in this present study were consistent with that reported for different genera of seaweeds collected elsewhere, globally (12–40% DW) [12,13,14]. In addition, protein content of *S. Polycystum* was within the range noted in the literature for brown seaweeds (3–15% DW) [14], even though the value for this brown seaweed species was lower when compared to that for red and green seaweed species (10–47% DW) [10]. Variations in the protein contents of seaweeds may be due to species differences and seasonal changes [15]. High content of protein (of up to 8.65%) in this brown seaweed could make it suitable as supplementary food, including animal feed and also, as a valuable resource to replace proteins from other sources. The measurement of low total lipids in *S. Polycystum* in this study recorded at 3.42% DW, is consistent with the fact already established in the literature that most seaweed species contain low contents of total lipid (<5% DW) [16,17]. This is reported to be attributed to the photosynthetic capabilities of seaweed pigments [18]. This notwithstanding, the values recorded in this study were higher than that reported (0.29% DW) by other researchers [10] for the same species. Nutritionally, lipid extracts of seaweeds naturally contain varieties of high-value polyunsaturated and monounsaturated fatty acids (FA) that act as catalysts for the antibacterial, antifungal, antiviral and cytotoxic properties of seaweeds [19]. The nutritional content of the *S. Polycystum* in terms of fibre and carbohydrate makes it a nutritive supplement which makes it an important human food resource. A study has reported that the fibre in seaweed helps to delay the feeling of hunger as it slows down stomach emptying, improves nutrient and mineral absorption thus, retarding hunger pangs [20]. These characteristics of seaweed fibre may be exploited in the design of diets to aid weight loss. Compared to terrestrial plant-derived foods, seaweed possesses similar or even higher amounts of dietary fibre. According to Rasmussen and Morrissey (2007) [21], seaweeds contain average total dietary fibre that could vary between 36% and 60% DW. Other studies have attributed the variations in the proximate compositions of *Sargassum* species measured at different locations to be related to seasonal differences of sample collection, taxonomic entities [22], geographical location of sampling [23] and environmental factors, including degree of salinity, intensity of light [24], temperature [25,26] and availability of nutrients [27,28].

Photosynthetic pigments are important to plant species and they are usually classified into three major categories: chlorophylls (*a*, *b*, *c*), carotenoids (carotene and xantophylls) and phycobilins (phycoerythrin and phycocyanin) [29]. Whereas chlorophylls comprise part of the components required for photosynthesis, the essential role played by carotenoids is to pass captured light energy to the chlorophylls. As photosynthetic pigments are vital components for organic synthesis of food in plants, cellular viability is thus associated with, and very much dependent on the photosynthetic activities of the seaweed [30]. Consequently, seaweeds distribute themselves in a vertical pattern on seashores and seafloors, an attribute that assists their maximum exposure to sun rays during low tides, as they get submerged under water in high tidal situations. Therefore, seaweeds experience continuous exposure to fluctuations in salinity, turbidity, high light intensity, pollution, temperature and heavy metal stresses, which are all harmful. The protection of these photosynthetic apparatuses against exposure to these stressors is of immense importance for seaweed survival [31]. It is worthy to note that the concentrations of specific photosynthetic pigments in seaweeds vary depending on the morphological structures of the species and the prevailing environmental factors [32]. A research carried out by Pangestuti and Kim, 2011 [29] related to natural pigments derived from marine algae, elucidated various beneficial biological activities to be considered, to select seaweed species for the cosmetics, food and pharmacology industries. Hence, it is no surprise that in the food industry, microalgae like spirulina is edible, as its contents of total carotenoid, chlorophyll *a* and chlorophyll *b* were measured as 3.5 mg g^−1^, 5.7 ± 0.2 mg g^−1^ and 3.4 ± 0.3 mg g^−1^, (on dry weight basis respectively [33] while *Chlorella* contained 81.81 µg g^−1^, 502 µg g^−1^ and 71.9 µg g^−1^ DW of total carotenoid, chlorophyll *a* and chlorophyll *b* respectively [34,35].

Nutritionally, diets containing high Na/K ratio are related to the incidence of hypertension, as the ideal ratio suggested is below 1.5. The ratio of Na/K measured in the present study (5.19) is relatively high when compared to previously published data (0.16) [10]. This may be attributable to variations in several factors, which include climate, location, waves, salinity, light exposure, pH, nitrogen availability, season, age of the species, metabolic processes and the affinity of the plant for each of these measured elements, among other likely reasons [36,37,38]. Seaweed, *Caulerpa lentillifera* (a green seaweed species) and olives (a species of plant) on the other hand, have been shown to have Na/K ratios measuring 7.8 and 45.63, respectively [10,39]. The recording of high calcium concentration in *S. polycystum* is of immense benefit, as it is essential for life. This is because Ca is one of the most important minerals that accumulate in seaweeds in much greater quantities than noted in terrestrial foodstuffs [40]. Additional to assisting in the protection and development of bone microstructure, Ca also enables muscle contraction, nerve conduction and blood coagulation. Magnesium on the other hand, is a vital element for bones and teeth. Ca and Mg, elements usually found in abundance in seaweeds, are also contained in apples, oranges, carrots and potatoes [41]. This is particularly crucial considering that there is current increasing preference for plant-based diets and lifestyles, leading to exclusion or reduction in the consumption of meat, eggs and dairy products [42]. Micro minerals such as iron (Fe), selenium (Se) and manganese (Mn) contribute significantly to the regular functioning of the human body, even though they are required in small quantities. Notably however, the Fe and Se content measured in this current study was relatively higher than that reported by other authors (68.21 and 1.14 mg 100/g DW) [10]. Mn content has been reported to be highest in red seaweed species, particularly in *Chondrus Crispus*, *Palmaria palmata* and *Gracilaria* spp. Those were sampled from different regions of Denmark [43]. Generally, the absorption potential of algae for trace metals is reported to occur via two mechanisms; first, is through a surface reaction that is independent of metabolic factors such as temperature, light, pH, or age of plants. This mechanism seems to be the primary absorption mechanism for Zn. The second process is a slower, active uptake, where metal ions such as Cu, Mn, Se and Ni are transported across the cell membrane into the cytoplasm. This second mechanism of absorption is directly dependent on metabolic processes and often differ with temperature, light or as the plant ages [38]. Consequently, the results for minerals content measured in the present study demonstrates that seaweeds species studied has potential to be utilised as a food supplement. This is especially as a source of Fe, to combat iron deficiency disorders in patients with anaemia; selenium, an antioxidant and catalyst for thyroid hormone production [44] and manganese for the formation of bones, connective tissues, sex hormones and blood clotting. Mn is also actively involved in fat and carbohydrate metabolism, absorption of calcium and the regulation of blood sugar, as well as for brain and nerve functions.

Major constituents of seaweed fibre are fucose, mannose, galactose and uronic acids, which vary based on the type of seaweed of interest [45]. These substances are cross-linked with each other to form complex structures of low to high molecular weight polysaccharides that are resistant to degradation but easily utilised by microorganisms [46]. The fibre component of seaweeds essentially contains structural polysaccharides such as alginate and fucoidan in brown seaweeds, carageenan, agar and porphyran rich in red seaweeds and ulvan found in green seaweeds. In its natural form, the seaweed plant is therefore, rich in polysaccharides, making it a suitable potential candidate for investigations as prebiotic. Accumulating evidence suggests that low molecular weight polysaccharides and oligosaccharides derived from hydrocolloids in seaweeds can act as potential sources for soluble fibre with prebiotic activity [46]. Studies suggest that polysaccharides and oligosaccharides extracted from seaweeds may simulate intestinal function, including fermentation, prevent pathogen adhesion and avoidance and potentially cure inflammatory bowel disease. Anticoagulant, antitumour, anti-inflammatory, antiviral, antihyperlipidemic or antioxidant functions are also shown in certain seaweed polysaccharides [47].

Generally, seaweeds are known to contain medicinally rich metabolites that include steroids, phenols, tannins, saponins, flavonoids, terpenoids and glycosides, which have been extensively studied and used in the pharmaceutical industry. These compounds were also found in abundance in other species of *Sargassum* such as *S. angustifolium*, *S. oligocystum* and *S. boveanum* [48]. In the present study, the presence of tannins was revealed in methanolic extracts of *S. polycystum*. Tannins have been found to exhibit antimicrobial properties as they are able to bind to adhesins and involved in enzyme inhibition, substrate deprivation and membrane disruption [49]. Saponins have such specific biological activities as anticancer, anti-inflammatory, antimicrobial and antioxidant properties [50,51]. Saponins also have the property to precipitate and coagulate red blood cells [52]. Flavonoids are hydroxylated phenolic substances implicated for their response to antioxidant activity [53]. The present study also demonstrated the presence of steroid in methanolic extracts of *S. polycystum*. Steroids are known to possess antimicrobial, anticancer, antiarthritic, antiasthma and anti-inflammatory properties and as such, they are considered as very important compounds [54]. Presence of terpenoids have been related to acquisition of cytotoxicity against a variety of cancer cells and cancer prophylaxis, while glycosides may be utilised as food additives and bio preservatives [55,56].

A previous study carried out on three seaweed species, using different physical pre-treatment strategies found that the protein pellet isolated from *P. palmate* contained the highest percentage of EAA, at 43.8%, followed by protein from *C. crispus* containing 40.94% EAA, while the lowest EAA was found in proteins from *F. vesiculosis*, at 25.45%. The EAAs found in protein of these three different seaweeds were obtained using classical protein extraction method [57]. The high content of the non-essential amino acids (NEAAs) aspartic and glutamine acids are known to be responsible for the special flavours and tastes of seaweeds [58]. In terms of human health, NEAAs are useful in nucleotide and lipid biosynthesis, maintenance of redox homeostasis and for various allosteric and epigenetic regulatory mechanisms, including other aspects of tumour metabolism. There is considerable interest to target NEAA metabolism for cancer therapy because of their importance in these different functions [59]. Generally, the content of total amount of amino acids in green seaweed is significantly higher than that in red and brown seaweeds, although this contrasts to reports by other researchers [60] on subtropical red and green seaweeds.

The most prevalent metabolite identified and measured in *S. polycystum* in this present study was n-hexadecanoic acid or palmitic acid C16:0, which is a form of fatty acid in plants. Palmitic acid has been reported to possess antioxidant properties, which displayed cytotoxicity against human colorectal carcinoma cells (HCT-116) [61,62]. This is in contrast to report from a previous study, which revealed linolenic acid, arachidonic acid and eicosapentaenoic acid as the highest metabolite in the same species of seaweed sampled from a different location [10]. Schmid et al., 2014 [63] suggested that besides differences between species, variations in fatty acid concentrations among seaweeds were mostly due to abiotic factors such as light, salinity and nutrients. Other metabolites identified in the present study were tetradecanoic acid, cis-vaccenic acid and 15-hydroxypentadecanoic acid. Additionally, 1,2-benzenedicarboxylic acid, mono(2-ethylhexyl) ester (BMEH) is known for its anti-fungal, anti-diabetic, anti-cancer, antioxidant activities and as a potent antimicrobial agent [64,65]. Selvakumar et al., 2019 [66] described that BMEH isolated from marine *Streptomyces* sp. VITJS4 demonstrated in vitro anti-cancer potential against liver (HepG2) cancer cells. Moreover, 3,7,11,15-tetramethyl-2-hexadecen-1-ol (Phytol) (also measured in the present study) is an important diterpene that possesses antimicrobial, antioxidant and anti-cancer activities [67,68]. The ability of phytol for anti-cancer activity may be associated with its ability to remove the hydroxyl radical (free radical) [69]. Benzenepropanoic acid, 3,5-bis(1,1-dimethylethyl)-4-hydroxy- methyl ester on the other hand, exhibits anti-fungal and antioxidant activities [70], while squalene which is vitamin E compound acquired for antidiabetic, anti-inflammatory and antioxidant activities. 

Physicochemical properties, including swelling capacity (SWC), water-holding capacity (WHC) and oil-holding capacity (OHC) of *S. polycystum* define the physiological effects of dietary fibre. It is vital to consider the water associated with fibre while searching the impact of fibre in the diet. The metabolic activity of fibre along the human gut is affected by water [71]. Generally, an increase is noticed in the SWC and WHC of *S. polycystum* powder with increasing temperature could be due to an increase in the solubility of the fibre component and the protein content of *S. polycystum* [72]. The SWC and WHC properties of seaweeds are usually related to their polysaccharide and protein characteristics that bind to the polysaccharide cell wall [73]. The differences noted in the seaweed samples with temperature in SWC and WHC may be due to protein conformations and the changes in the number and nature of the protein molecules’ water-binding sites [60]. For food industry applications, high WHC values suggest that the seaweed material may be useful as functional component for modifying the texture of meat products and salad dressings to develop low caloric foods. Low-caloric sugar substitutes are also essential for low-calorie food products such as extruded chips, maize flakes, cookies and crackers [74,75]. The observation in the current study that the OHC of *S. polycystum* at room temperature was lower than that at 37 °C and 60 °C, but slightly higher at 80 °C is a desirable attribute in the food industry. OHC characterises the emulsification properties of *S. polycystum*, where high OHC value in an ingredient has been shown to allow for the stabilisation of food emulsions in high-fat food products [76]. It is an important functional property of food ingredients that is measured to determine the hydrophobicity of fibre molecules [77].

## 4. Materials and Methods

### 4.1. Sample Collection and Preparation

Fresh brown seaweed, *S. polycystum* was collected in the morning, by hand during low tides on reef flats, approximately 100 m from the shore in Teluk Kemang, Port Dickson, Malaysia (2°26′ N, 101°51′ E) in September 2019. The collected seaweed sample was cleaned of extraneous materials like epiphytes, sand particles, pebbles and shells by washing with sea water. The samples were separated into two. The first portion of the samples was taken into the herbarium for taxonomic identification. The taxonomic identification was crossed reference with taxonomic books, monographs and reference herbaria. The second portion of clean samples was then placed in transparent polyethylene bags (inside chilled plastic containers) and transported to the research facilities of the Aquatic Animal Health and Therapeutics Laboratory (AquaHealth), Institute of Bioscience, Universiti Putra Malaysia (UPM) Serdang, Malaysia. In the laboratory, the sample was further washed thoroughly with tap water, followed by distilled water and frozen over- night at −80 °C in a freezer (Thermo Scientific, Asheville, NC, USA). Frozen seaweed sample was lyophilised in a freeze dryer (Labconco FreeZone, Kansas, USA) until a constant weight of biomass was attained. The dried seaweed sample was subsequently ground to a fine powder with a laboratory-scale blender and sieved using a 200 micron-sized sieve. The fine sample powder was collected in screw-capped bottles, labelled appropriately and stored in a freezer maintained at −80 °C until further use.

### 4.2. Proximate Composition Determination

Dry matter, ash, crude fat (by ether extraction), crude protein and total dietary fibre contents were analysed using official methods of AOAC (2006) [78]. The moisture content was determined by oven-drying method, by drying 1 g of sample at 105 °C until constant weight (AOAC 950.46) in a vacuum oven (Memmert ULM 400, Schwabach, Germany) and the ash content was gravimetrically determined by heating the sample at 550 °C for 4 h in a muffle furnace (Carbolite 11/14, Hope Valley, UK) (AOAC 923.03). The fat content was extracted in a Soxtec system with petroleum ether (AOAC 991.36) and the protein content was determined using Kjeltec system (N x 6.25) (AOAC 981.10). Crude fibre was determined with successive hydrolysis with 100 °C 0.26 N sulphuric acid (H_2_SO_4_) and 0.31 N sodium hydroxide (NaOH) for 30 min each in a digital hot plate (AOAC 962.09). Dietary fibre was analysed with enzymatic-gravimetric method (AOAC 991.43). NFE content was calculated by difference;
NFE = 100 − (crude protein + lipids + ash + crude fibre)

The results were expressed on dry weight (DW) basis and all measurements were performed in triplicate. 

### 4.3. Mineral Determination

In the determination of mineral elements in *S. polycystum*, 1 g of sample was dissolved in 1 mL of nitric acid (HNO_3_) and hydrogen peroxide (H_2_O_2_) before it was digested in microwave. The mixture was shaken and filtered using filter paper. The minerals: potassium (K), magnesium (Mg), calcium (Ca), iron (Fe) and manganese (Mn) were determined using an atomic absorption spectrophotometer (AAS) (Hitachi Z-5000, Tokyo, Japan) and an air-acetylene burner was used. Sodium (Na), phosphorus (P) and selenium (Se) were determined using inductive coupled plasma mass spectrometry (ICP-MS) (Perkin Elmer ELAN 9000, Wellesley, MA, USA). The concentrations of the elements were determined from calibration curves of the standard elements. The results were expressed in mg 100/g DW basis and all measurements were performed in triplicate.

### 4.4. Total Carotenoids and Chlorophyll Content Determination

Total carotenoids, chlorophylls *a* and *b* contents of dried *S. polycystum* were determined in accordance with slight modifications to the method of Lichtenthaler and Buschmann, (2001) [79]. According to this method, dilution in methanol was carried out for seaweed lipophilic extract and the absorbance of total carotenoids; chlorophyll *a* and chlorophyll *b* were measured at 470, 665.2 and 652.4 nm, respectively using a UV-spectrophotometer (Shimadzu UV 1601, Kyoto, Japan) Calculation of the total carotenoids, chlorophyll *a* and chlorophyll *b* contents in seaweed lipid sample were performed using the Lichtenthaler equations as follows (with values expressed as µg/g dry weight [µg/g DW]);
C(_x+c_) (µg/mL) = (1000 A_470_ − 1.63 C_a_ − 104.96 C_b_)/221
C_a_ (µg/mL) = 16.72 A_665.2_ − 9.16 A_652.4_
C_b_ (µg/mL) = 34.09 A_652.4_ − 15.28 A_665.2_
where, C_(x+C)_ = total carotenoids, C_a_ = chlorophyll *a*, C_b_ = chlorophyll *b*, A_470_ = absorbance at 470 nm, A_665.2_ = absorbance at 665.2 nm, A_652.4_ = absorbance at 652.4 nm.

### 4.5. Amino Acid Determination

The amino acid analysis of *S. polycystum* was performed using high performance liquid chromatography (HPLC). The amino acids (AAs) content of *S. polycystum* was determined according to Chan and Matanjun, (2017) [76] with some slight modification. Powder of *S. Polycystum* (0.30 g) was weighed in glass stoppered test tube and hydrolysed with 5 mL of 6 N hydrochloride acid (HCl) at 110 °C for 24 h. The hydrolysates were then cooled to room temperature and quantitatively transferred into 100 mL volumetric flask, with 400 μL of 50 μmol/mL α-aminobutyric acid (AABA) in 0.1 M HCl added with ultra-pure water. The aliquot was filtered through filter paper before filtering again through a 0.2 µm nylon syringe filter. Later, 10 μL filtered solution was collected into a microcentrifuge tube for derivatisation process. About 10 μL of samples and standards were injected into a HPLC (Waters 2475, Waters Co., Milford, MA, USA) with the flow rate set at 1 mL/min. The mobile phase used was: (A) AccQ Tag Eluent A (200 mL AccQ Tag to 2 L of ultrapure water) and (B) HPLC grade acetonitrile (60%). The mobile phase was filtered through 0.45 µm cellulose membrane filters before it was used. All separations were carried out with AccQ Tag column (3.9 × 150 mm, particle size 4 µm). Detection was carried out by a fluorescence detector operated with a 250 nm excitation and a 395 nm emission wavelength. The linear gradient condition was set as follows: 100% A at start, 2% B at 0.5 min, 9% B at 15 min, 13% B at 19 min, 35% B at 33 min, 35% B at 35 min, 100% B at 36 min, 100% B at 39 min, 100% A at 40 min and 100% A at 50 min. The quantity of each AA was determined from the peak area of known quantity of AA standard mixture and peak area of individual AA in sample that contained internal standard. The amount of each AA in *S. polycystum* was expressed as mg/g DW basis. All the measurements of AAs were performed in triplicate. The AA score of EAAs was calculated using the following equation:AA score (%) = (mg EAA in 1 g of test protein)/(mg EAA in 1 g of reference protein)*100

### 4.6. Metabolite Profile Using Gas Chromatography-Mass Spectrometry Analysis

Metabolite profile of *S. polycystum* was analysed by gas chromatography-mass spectrometry (GC-MS). For the metabolite extraction from *S. polycystum*, 1 g of powdered samples was dissolved with 50 mL methanol, and vortexed for 5 min. This solution was extracted by sonication for 30 min using ultrasonic water bath (Power Sonic 505, Seoul, Korea) at ambient temperature. The solvent extract was then filtered through Whatmann No. 1 filter paper while the dry residue re-extracted with another 50 mL of methanol for a second round of extraction. The extraction process was subsequently repeated in the same manner for a third round of extraction. The filtered extracts were pooled and evaporated to dryness using a rotary evaporator (N-1001S-WD, with EYELA Oil Bath OSB-2000, Tokyo, Japan) at 30 °C, and stored at −20 °C until further analysis. Methalonic extracts were characterised quantitatively via GC-MS, with slight modifications [80] using a Shimadzu QP2010 Plus GC-MS system. In the experimental procedure, 0.5 μL of sample was separated on a Zebron ZB5-ms 30 m × 0.25 mm ID × 0.25 μm film thickness) column. Splitless injection was performed using a purge time of 1 min. Helium represented the carrier gas at a flow rate of 1 mL/min. The column temperature was maintained at 50 °C for 3 min, then programmed at 250 °C for 10 min and maintained at 250 °C for 30 min. The inlet and the detector temperatures were set at 250 °C and the solvent cut time was set at 4.50 min. Identification of peaks was based on a computer-based program matching the mass spectra with those in the library for National Institute of Standards and Technology (NlST3208 and NIST 08s). This was done by comparing retention time data with that obtained for authentic laboratory standards. Individual detected peak areas were quantified and expressed as percentage of total components detected.

### 4.7. Colorimetric Determination of Monosaccharide in S. polycystum

To determine the concentration of fucose from extracts the method previously described by Dische (1955) [81] was employed. Briefly 4.5 mL of diluted H_2_SO_4_ with distilled water (6:1 *v*/*v*) was added to 1 mL of dilute samples in test tubes. These tubes were placed on ice and allowed to cool for 1 min while stirring with a glass rod. The samples tubes were boiled in water bath for 10 min then cooled with tap water. The absorbance was read at 396 and 427 nm on the spectrophotometer (Shimadzu UV-1601 UV-VIS visible, Kyoto, Japan). Cysteine solution (0.1 mL) was added to the tubes and left to incubate for 30 min at room temperature then absorbance was read at 396 and 427 nm once again. The absorbance due to fucose is the difference between reading at 396 and 427 nm. Differences in these measurements (after subtraction from pre-cysteine addition absorbance) were used to directly correlate methyl pentose (L-fucose, L-mannose, glucuronic acid, rhamnose, xylose and galactose) concentration by using a standard curve obtained with a fucose concentration ranging from 5–100 µg/mL. Total fucose concentration in mg, was calculated from the formula: F = b × 0.06 × V,
whereF = Quantity of fucose in mgb = the difference in absorbance readingsV = the total volume in mL0.06 = factor converting absorbance to amount of fucose

### 4.8. Phytochemical Screening

Phytochemical examination was carried out for *S. polycystum* according to standard methods. The dried seaweed was ground into a fine powder and dissolved with 50 mL methanol, and vortexed for 5 min. This solution was extracted by sonication for 30 min using ultrasonic water bath (Power Sonic 505, Seoul, Korea) at room temperature. The solvent extract was then filtered through Whatmann No. 1 filter paper while the dried residue re-extracted with another 50 mL of methanol for a second round of extraction. The extraction process was subsequently repeated in the same manner for a third round of extraction. The filtered extracts were pooled and evaporated to dryness using a rotary evaporator (N-1001S-WD, with EYELA Oil Bath OSB-2000, Tokyo, Japan) at 30 °C, and stored at −20 °C until further analysis. For terpenoids, extract was dissolved in 2 mL of chloroform and evaporated to dryness. To this, 2 mL of concentrated sulphuric acid, H_2_SO_4_ was added and heated for about 2 min. A greyish colour indicated the presence of terpenoids. The presence of steroids was indicated by the development of a greenish colouration by mixing extract with 2 mL of chloroform. Then 2 mL each of concentrated H_2_SO_4_ and acetic acid was poured into the mixture. While for test for phenols and tannins, extract was mixed with 2 mL of 2% FeCl_3_ solution. A blue-green or black colouration indicated the presence of phenols and tannins. Detection for flavonoids was conducted by diluting extract with 2 mL of sodium hydroxide, NaOH. Mixture turned to intense yellow colouration. Once hydrochloric acid, HCl was added, the solution became colourless. As for saponins, extract was mixed with 5 mL of distilled water in a test tube. It was shaken vigorously. Formation of stable foam was taken as an indication for the presence of saponins.

### 4.9. Physicochemical Properties

Physicochemical properties such as swelling, water retention and oil-holding capacity were assessed in *S. polycystum* using the previously published method [76]. All measurements were performed in triplicate. For water swelling capacity (SWC), 0.50 g of lyophilised *S. polycystum* powder was mixed with 10 mL distilled water and stirred in a measuring cylinder before it was left at RT (25 °C), 37 °C, 60 °C and 80 °C for 18 h. The swelling volume was measured and expressed as mL of sample occupied per g DW of sample. The water-holding capacity (WHC) of sample was determined by dispersal of 0.50 g of lyophilised *S. polycystum* in 25 mL of distilled water and placed in a pre-weighed centrifuge tube. The dispersion was then stirred and left at RT (25 °C), 37 °C, 60 °C and 80 °C for 1 h before the mixture was centrifuged at 3000 g for 25 min. The supernatant was discarded, and the moisture content of sample was determined by dehydration in an oven at 50 °C for 24 h. The results were expressed as g/g DW of sample. The oil-holding capacity (OHC) of lyophilised *S. polycystum* was determined by weighing 0.5 g of sample and mixed with 20 mL of corn oil in a pre-weighed centrifuge tube. The mixture was then stirred before it was left at RT (25 °C), 37 °C, 60 °C and 80 °C concurrently for 1 h prior to centrifuging the mixture at 3000× *g* for 25 min. The oil supernatant was discarded and measured. The results were expressed as g/g DW of sample.

## 5. Conclusions

In conclusion, *S. polycystum* seaweed is a potential source for many bioactive, nutritionally and physiologically important compounds that could be utilised in the nutraceutical and pharmaceutical industries as sources of food and medicine, respectively. The present study demonstrated that the brown seaweed, *S. polycystum* is rich in proteins, lipids, carbohydrate and minerals such as Na, K, Ca, Mg Fe, Se and Mn. From results of the phytochemical investigation of *S. polycystum* extract, the presence was revealed of various secondary metabolites, in which it was discovered that n-hexadecanoic acid, 1,2-benzenedicarboxylic acid, mono(2-ethylhexyl) ester, benzenepropanoic acid, 3,5-bis(1,1-dimethylethyl)-4-hydroxy- methyl ester, 1-dodecanol, 3,7,11-trimethyl-, were the most abundant metabolites. The present study also demonstrated the content of pigments (total carotenoids, chlorophylls *a* and *b*) in *S. polycystum* and that the major constituent of *S. polycystum* fibre included fucose, mannose, galactose, xylose and rhamnose. The physicochemical properties of this seaweed, which included water-holding and swelling capacity were comparable to that in some commercial fibre-rich products. Results of this study therefore, indicate that *S. polycystum* is a potential candidate as a functional food source for human consumption and its cultivation needs to be encouraged.

## Figures and Tables

**Table 1 molecules-26-05216-t001:** Proximate composition of ground lyophilised *S. polycystum*.

Composition	% DW
Moisture	13.70 ± 0.14
Ash	21.38 ± 0.17
Protein	8.65 ± 1.06
Lipid	3.42 ± 0.01
Crude fibre	13.55 ± 0.14
Carbohydrate	36.55 ± 1.09
Total Dietary fibre	2.75 ± 0.58

Values are mean ± SEM, *n* = 3. Dry weight (DW).

**Table 2 molecules-26-05216-t002:** Content of total carotenoids, chlorophyll *a* and chlorophyll *b* in extracts of s elected seaweed species sampled in the Malaysian Peninsular.

Seaweed Species	Total Carotenoids (µg/100 g DW)	Chlorophyll *a* (µg/100 g DW)	Chlorophyll *b* (µg/100 g DW)
*S. polycystum*	45.28 ± 1.77	141.98 ± 1.18	111.29 ± 2.28

Values are expressed as mean ± SEM of three (3) replicate measurements (*n* = 3), dry weight basis.

**Table 3 molecules-26-05216-t003:** Mineral content of brown seaweed, *S. polycystum*.

Minerals	mg/100g DW
Na	8876.45 ± 0.47
K	1711.05 ± 0.07
Mg	213.85 ± 0.02
Ca	1079.75 ± 0.30
Fe	277.60 ± 0.12
Se	4.70 ± 0.01
Mn	4.45 ± 0.01
P	19,108.25 ± 0.70
Na/K ratio	5.19 ± 2.59
Total cation	31,276.10 ± 1.67

Values are mean ± SEM, *n* = 3. Dry weight (DW); sodium (Na); potassium (K); magnesium. (Mg); calcium (Ca); iron (Fe); selenium (Se); manganese (Mn); phosphorus (P).

**Table 4 molecules-26-05216-t004:** Monosaccharides composition of fucose-containing sulphated polysaccharides in *S. polycystum* (µg/mL).

Monosacharide	Composition (µg/mL)
Fucose	23.00 ± 0.02
Mannose	0.45 ± 0.02
Galactose	0.75 ± 0.23
Xylose	1.00 ± 0.04
Rhamnose	0.30 ± 0.17

Values are mean ± SEM, *n* = 3.

**Table 5 molecules-26-05216-t005:** Phytochemicals from *S. polycystum*.

Metabolites	Steroid	Phenol	Tannins	Saponins	Flavonoids	Terpenoids	Glycosides
*S. polycystum*	+	+	+	+	+	+	+

(+) presence, of tested metabolite.

**Table 6 molecules-26-05216-t006:** Amino acid content of *S. polycystum*.

Amino Acid	mg/g DW
Arginine, Arg ^a^	4.32 ± 0.14
Thyrosine, Tyr ^a^	1.55 ± 0.03
Threonine, Thr ^a^	3.72 ± 0.03
Valine, Val ^a^	5.58 ± 0.13
Methionine, Met ^a^	1.25 ± 0.01
Lysine, Lys ^a^	3.7 ± 0.02
Isoleusine, Ile ^a^	4.63 ± 0.10
Leucine, Leu ^a^	7.57 ± 0.19
Phenylalanine, Phe	4.95 ± 0.13
Alanine, Ala	5.78 ± 0.14
Aspartic, acid, Asp	7.47 ± 0.14
Glutamic acid, Glu	10.01 ± 0.20
Serine, Ser	4.96 ± 0.13
Glycine, Gly	5.43 ± 0.06
Proline, Pro	3.94 ± 0.05
Total AAs	74.90 ± 1.45
Total EAAs	37.28 ± 0.78
Total NEAAs	37.63 ± 0.74
EAAs/Total AAs	0.50 ± 0.54
EAAs/NEAAs	1.00 ± 1.05

Values are mean ± SEM, *n* = 3. Dry weight (DW); amino acids (AAs); essential amino acids (EAAs); non-essential amino acids (NEAAs). ^a^ Essential amino acid.

**Table 7 molecules-26-05216-t007:** Metabolite profile of *S. Polycystum*.

No	Name	Retention Time (min)	Area%	Molecular Formula
	***Phenol***			
1	Phenol, 3,5-bis(1,1-dimethylethyl)-	15.983	3.62	C_14_H_22_O
	***Ketone***			
2	2-Pentadecanone, 6,10,14-trimethyl-	19.698	0.69	C_18_H_36_O
	***Aldehyde***			
3	*E*-14-Hexadecenal	16.904	1.01	C_16_H_30_O
4	Octadecanal	17.937	0.43	C_18_H_36_O
	***Alcohol***			
5	3,7,11,15-Tetramethyl-2-hexadecen-1-ol	20.067	2.27	C_20_H_40_O
6	1-Dodecanol, 3,7,11-trimethyl-	22.341	4.26	C_15_H_32_O
	***Hydrocarbons***			
7	1-Hexadecene	14.384	0.51	C_14_H_28_
8	1-Octadecene	19.161	0.83	C_18_H_36_
9	1-Nonadecene	21.206	0.30	C_19_H_38_
	***Fatty acids***			
10	Tetradecanoic acid	18.913	1.74	C_14_H_28_O_2_
11	*n*-Hexadecanoic acid	21.027	10.35	C_16_H_32_O_2_
12	cis-Vaccenic acid	22.692	3.37	C_18_H_34_O_2_
13	15-Hydroxypentadecanoic acid	24.005	1.49	C_15_H_30_O_3_
	***Sterols***			
14	Stigmasta-5,22-dien-3-ol, acetate, (3*β*)-	30.024	0.83	C_31_H_50_O_2_
15	Cholest-4-en-3-one	31.086	0.87	C_27_H_44_O
16	Stigmasta-4,24(28)-dien-3-one, (24*E*)-	32.357	1.52	C_29_H_46_O
	***Fatty acid methyl/ethyl esters***			
17	Tridecanoic acid, 12-methyl-, methyl ester	18.427	0.53	C_15_H_30_O_2_
18	Hexadecanoic acid, 2,3-dihydroxypropyl ester	25.319	0.66	C_19_H_38_O_4_
	***Other***			
19	Benzenepropanoic acid, 3,5-bis(1,1-dimethylethyl)-4-hydroxy-, methyl ester	20.645	7.03	C_18_H_28_O_3_
20	Hexadecanoic acid, 2-hydroxy-1-(hydroxymethyl)ethyl ester	23.601	1.17	C_19_H_38_O_4_
21	1,2-Benzenedicarboxylic acid, mono(2-ethylhexyl) ester	25.946	7.69	C_16_H_22_O_4_
22	Squalene	27.992	0.74	C_30_H_50_

**Table 8 molecules-26-05216-t008:** The SWC, WHC and OHC of *S. polycystum* at various temperatures.

Parameters	RT	37 °C	60 °C	80 °C
SWC (mL/g DW)	10.27 ± 0.25	10.43 ± 0.12	10.17 ± 0.29	10.27 ± 0.06
WHC (g/g DW)	2.76 ± 1.02	4.10 ± 0.84	2.89 ± 0.86	2.90 ± 0.69
OHC (g/g DW)	1.90 ± 0.23	2.04 ± 0.30	2.27 ± 0.05	1.72 ± 0.19

Values are mean ± SEM, *n* = 3. RT: room temperature; dry weight (DW); swelling capacity (SWC); water-holding capacity (WHC); oil-holding capacity (OHC).

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
