# Peer review of "Chemical, Nutrient and Physicochemical Properties of Brown Seaweed, Sargassum polycystum C. Agardh (Phaeophyceae) Collected from Port Dickson, Peninsular Malaysia"

_molecules, 2021, doi:10.3390/molecules26175216_

Round 1
Reviewer 1 Report
This manuscript deals with an interesting and original study on the potentials of seaweed species in coast of Peninsular Malaysia for utilization as functional food for human consumption, in order to promote for the commercial cultivation of the species.
The paper is clearly presented and results are very useful. However, I have some suggestions:
- Please, revise all table mean and standard deviation: the number of digits of the mean value depends on the place where the significant digit appears and the number of digits of the corresponding data should be adjusted by taking into account the corresponding standard deviation values. In this way, each value in the Tables has been expressed with the significant digits according to the significant digits of each standard deviation value. Correct the different errors in the tables, keeping in mind that 0 and 1 are not significant digits.
- Lines 256-259. Please compare the values of carotenoids and chlorophylls to values of Chlorella, not only Spirulina. There are recent works where algae are used as ingredients and these pigments are studied. Chrorella is an important microalga such as Spirulina.
Author Response
The paper is clearly presented and results are very useful. However, I have some suggestions:
- Please, revise all table mean and standard deviation: the number of digits of the mean value depends on the place where the significant digit appears and the number of digits of the corresponding data should be adjusted by taking into account the corresponding standard deviation values. In this way, each value in the Tables has been expressed with the significant digits according to the significant digits of each standard deviation value. Correct the different errors in the tables, keeping in mind that 0 and 1 are not significant digits.
Response:
The observation by the reviewer on the differences in the number of decimal digits of data for results on different Tables, is graciously acknowledged. In keeping to the suggestion, notice that data presented in ALL Tables and as referenced in the manuscript are now reported consistently to 2 places of decimal (except for the Retention time, RT on Table 7). Retention times refer to the time (in minutes, seconds and milliseconds) for the appearance of metabolite peaks, which are recorded directly according to results from GC-MS analysis, and cannot be rounded-off.
- Lines 256-259. Please compare the values of carotenoids and chlorophylls to values of Chlorella, not only Spirulina. There are recent works where algae are used as ingredients and these pigments are studied. Chrorella is an important microalga such as Spirulina.
Response:
Authors are in agreement that the observation by the reviewer has merit, given that both Spirulina and Chorella are micro-algae. However, the authors decided to make comparison in the values for chlorophylls between the brown seaweed species to Spirulina and NOT to that for Chlorella because Chlorella is a single-celled micro-algae, whereas Spirulina is a multi-cellular, filamentous micro-algae. Appearance of Spirulina as filaments makes it relatively large in size during cultivation to be compared to the brown seaweed, which is a macro-algae, as opposed to Chlorella, which appear as small, almost microscopic, single-celled organisms. However, the value of the carotenoid and chlorophyll a and b value in Chlorella has been included in the manuscript.

Reviewer 2 Report
Still a point should be addressed.
As mentioned in the cover letter, the samples were collected from the sampling location at the peak period of their bloom. What's the month? Please describe in the "4.1. Sample collection and preparation" part.
Author Response
As mentioned in the cover letter, the samples were collected from the sampling location at the peak period of their bloom. What's the month? Please describe in the "4.1. Sample collection and preparation" part.
Response:
As was suggested by the reviewer, the amendment recommended has been made accordingly, as the month and year of sampling are now included.

Reviewer 3 Report
The authors introduced only slight corrections to the text to the Reviewer's suggestion. The work contains significant errors, requires rethinking and redrafting.
The authors declared that the mistakes had been fixed. Unfortunately, this is not the case. Some examples below:
Lines: 88-89: The authors report that the total carbohydrate content is 55.58%. There is no such value in table 1.
Line: 116. "Brown" seaweed or "brown" seaweed. Once written with a capital letter and once with a lower case (e.g. line: 84). Please standardize.
Line: 149. "Amino" acids, please change to "amino".
Table 8. Table title. Is it SC or SWC?
Below tables 1-4 and 6 and 8, the following information: Values ​​are "Mean" - I suggest replacing it with a lower case letter "mean".
Lines: 211-241. The authors state: "protein content of S. Polycystum was within the range noted in the literature for brown seaweeds (3–15% DW), even though the value for this brown seaweed species was lower when compared to that for red and green seaweed species (10-47% DW).
And in line 216, the authors state: "Higher protein content in seaweed could make it suitable as supplementary food ...." So what is the higher protein content?
Was only one test performed for the data in Table 7? Why is there no information under the table? Meanwhile, in the Materials and Methods section, the authors' quote (lines: 426-427): "all measurements were performed in triplicate".
Table 6. The authors should indicate which amino acids are essential. It is not enough to list the sum of essential and other amino acids.
The discussion of the thesis remained unchanged, despite the Reviewer's suggestion. The discussion of the results must be related to the purpose of the work.
Lines: 228-230
The authors state: "A study has reported that the fiber in seaweed helps to delay the feeling of hunger as it slows down stomach emptying [20], thus aiding weight loss" The conclusion is too far-fetched ('thus aiding weight loss').
Author Response
The authors introduced only slight corrections to the text to the Reviewer's suggestion. The work contains significant errors, requires rethinking and redrafting.
The authors declared that the mistakes had been fixed. Unfortunately, this is not the case. Some examples below:
Lines: 88-89: The authors report that the total carbohydrate content is 55.58%. There is no such value in table 1.
Response:
We have corrected this inadvertent mistake, which may be noticed to have been effected in the Table in the previous editing, but was not corrected in both the Abstract and the main body of the manuscript.
Line: 116. "Brown" seaweed or "brown" seaweed. Once written with a capital letter and once with a lower case (e.g. line: 84). Please standardize.
Response:
We have corrected this inadvertent mistake, for consistency.
Line: 149. "Amino" acids, please change to "amino".
Response:
We have corrected this inadvertent mistake.
Table 8. Table title. Is it SC or SWC?
Response:
We have corrected this inadvertent mistake.
Below tables 1-4 and 6 and 8, the following information: Values ​​are "Mean" - I suggest replacing it with a lower case letter "mean".
Response:
As was suggested by the reviewer, the amendments recommended have been made accordingly
Lines: 211-241. The authors state: "protein content of S. Polycystum was within the range noted in the literature for brown seaweeds (3–15% DW), even though the value for this brown seaweed species was lower when compared to that for red and green seaweed species (10-47% DW).
And in line 216, the authors state: "Higher protein content in seaweed could make it suitable as supplementary food ...." So what is the higher protein content?
Response:
On Lines 211-241, the statement; “protein content of S. Polycystum was within the range noted in the literature for brown seaweeds (3–15% DW), even though the value for this brown seaweed species was lower when compared to that for red and green seaweed species (10-47% DW)” was to justify that the result (8.65%) we noted in the evaluation we carried out on the brown seaweed species was consistent with that noted in the literature for brown seaweed species (of 3-15%). But in comparison to the protein content noted in the literature (of between 10-47%) for red and green seaweed species however, the value of our brown species of 8.65% was comparatively lower. This was just to establish facts.
The above statements of comparison have merit, considering the fact that the majority of the traditionally known “edible seaweeds” are green and red seaweeds. Majority of brown seaweeds reported in the literature however, are shown to contain low (3-5%, although with a few others containing up to 15%) proteins and are mostly recommended for their pharmaceutical values, based on their profile of important phytochemicals. That we evaluated this brown species and noted relatively high (8.65%) proteins, accounted for the second statement on line 216. However, as the reviewer has noted ambiguity in the earlier reportage, the statement has now been edited, to avoid this mix-up.
Was only one test performed for the data in Table 7? Why is there no information under the table? Meanwhile, in the Materials and Methods section, the authors' quote (lines: 426-427): "all measurements were performed in triplicate".
Response:
The evaluation for the metabolite profile, the result of which is shown in Table 7 was performed as single test. This is as the practice for GC-MS evaluations is based on comparison of identified peaks (by a computer programme) with those of standards available in an International library. As this is already reported in the methodology narrative, it is usually not indicated anywhere on the Table, in keeping with the usual practices for such results.
In the Materials and Methods section, (lines: 426-427): the designation suggesting "all measurements were performed in triplicate" was in reference to that for the proximate composition determination. This is the reason why the value of n = 3 was shown as foot note on Table 1.
Table 6. The authors should indicate which amino acids are essential. It is not enough to list the sum of essential and other amino acids.
Response:
As was suggested by the reviewer, the amendment recommended has been made accordingly. Essential amino acids are now identified with a superscript lowercase alphabet on the Table and defined at the foot note, for ease of identification.
The discussion of the thesis remained unchanged, despite the Reviewer's suggestion. The discussion of the results must be related to the purpose of the work.
Response:
As observed by the reviewer, “the discussion of the thesis (has) remained unchanged” because authors have gone thoroughly through the manuscript again and noted that discussions of the results are related to the purpose of the work, as was suggested. Authors wish to call the reviewer’s attention to the purpose of the work, which are shown in both the Abstract and Introduction to be “to explore potential application of the seaweed species abundant on the coast of Peninsular Malaysia, for utilization as functional food, in order to promote the commercial cultivation of the species.” In doing this, authors note that most discussions of results in the manuscript were in relation to application in the food industry or as functional food component of the phytochemicals noted. The suggestion by the reviewer has largely been graciously taken into account, leading to the conclusion of the manuscript.
Lines: 228-230 The authors state: "A study has reported that the fiber in seaweed helps to delay the feeling of hunger as it slows down stomach emptying [20], thus aiding weight loss" The conclusion is too far-fetched ('thus aiding weight loss').
Response:
The quoted statement is attributed to the potential benefits of seaweed fibre in the food industry, where documented benefits were cited and their potential use mentioned. However, as observed by the reviewer, authors agreed that the conclusion to that statement as earlier stated, seemed too far-fetched. To put this into proper context, we have now cited the full benefits appropriately and provided the potential use it portends for the food industry. The manuscript has thus been edited to reflect this.

Round 2
Reviewer 3 Report
Thank you the authors for the changes made.
This manuscript is a resubmission of an earlier submission. The following is a list of the peer review reports and author responses from that submission.
Round 1
Reviewer 1 Report
General attention.
The work requires a thorough change, redrafting.
It also contains linguistic errors, so the work should be carefully checked.
Selected detailed comments:
Abstract - the authors only provide results on selected ingredients of freeze-dried brown seaweed, Sargassum polycystum.
Meanwhile, the aim of the work (lines: 68-73) was: "to evaluate the chemical composition, identify metabolites and physicochemical properties of freeze-dried brown seaweed, S. polycystum from Port Dickson, Malaysia. The overall aim of the study was to explore the potentials of the seaweed species in coast of Peninsular Malaysia for utilization as a functional food for human consumption, in order to promote for the commercial cultivation of the species ".
So abstract does not relate to the purpose of the work.
Introduction - the purpose of the work was not explained.
Results:
In Table 1, there were summarized results of the proximate composition of brown seaweed, S. polycystum. Why is the composition greater than 100%. Is there a mistake in the research methodology?
Line: 339 - what was the size of the sample taken for the research. Was the sample divided into two parts of the same size?
Discussion
The discussion of the results should be thoughtful and re-edited. It is chaotic in its present form.
The discussion cannot only be about comparing the values ​​of the selected ingredients without drawing conclusions.
Line: 162-164: The authors' quote: Seaweeds, thus, generally vary from terrestrial plants in their morphological and physiological characteristics, as well as their chemical compositions, as is evident in their different appearances.
And then, on lines 166-167, authors say that "The ash content was higher than most land-based plants, with ash values ​​ranging from 5 to 10%."
So why do the authors compare these results?
Reviewer 2 Report
The manuscript described the determination of chemical compounds and some physicochemical properties of a brown seaweed of Sargassum polycystum. Fruitful results were given in this manuscript, while some points should be addressed.
1. The study material is Sargassum polycystum. What is the reason for study this seaweed? The annual production or the significance of the material should be given. Thus the purpose of this study will be clear.
2. The chemical compositions strongly rely on harvest period and position of seaweed. Why the authors haven't compared different samples?
3. Line 66: only few studied on the bioavailability of nutrients and phytochemicals from algal foods. Why did the author list this fact here? In this manuscript, the authors didn't study the bioavailability of this seaweed either.
4. The novelty or the similarity of chemical compositions of Sargassum polycystum should be compared with those of other commonly harvested Sargassum seaweed, for expamle, Sargassum fusiformis.
5. "Results of this study indicate that S. polycystum are the potential candidates for cultivation as functional food sources for human consumption." This conclusion is not sufficient based on the research data, because only chemical compositions and some basic physicochemical properties have been carried out. A more exact conclusion should be provided.
Reviewer 3 Report
This manuscript deals with an interesting and original study on the potentials of seaweed species in coast of Peninsular Malaysia for utilization as functional food for human consumption, in order to promote for the commercial cultivation of the species.
The paper is clearly presented and results are very useful. However, I have some suggestions:
- Please, revise all table mean and standard deviation: the number of digits of the mean value depends on the place where the significant digit appears and the number of digits of the corresponding data should be adjusted by taking into account the corresponding standard deviation values. In this way, each value in the Tables has been expressed with the significant digits according to the significant digits of each standard deviation value. Correct the different errors in the tables, keeping in mind that 0 and 1 are not significant digits.
- Table 2. Use µg/100 g DW as units.
- Table 3. Change mg100/g DW by mg/100g DW.
- Lines 195-213. Please compare the values of carotenoids and chlorophylls to values of other edible algae such as Chlorella or Spirulina. There are recent works where algae are used as ingredients and this pigments are studied.